# CLoQ: Enhancing Fine-Tuning of Quantized LLMs via Calibrated LoRA Initialization

**Yanxia Deng**[*]                                                                                   *ydeng5@albany.edu*
*Department of Mathematics and Statistics*
*University at Albany, SUNY*

**Aozhong Zhang**[*]                                                                                 *azhang3@albany.edu*
*Department of Mathematics and Statistics*
*University at Albany, SUNY*

**Selcuk Gurses**                                                                                   *sgurses@albany.edu*
*Department of Mathematics and Statistics*
*University at Albany, SUNY*

**Naigang Wang**                                                                                    *nwang@us.ibm.com*
*IBM T. J. Watson Research Center*

**Zi Yang**                                                                                         *zyang8@albany.edu*
*Department of Mathematics and Statistics*
*University at Albany, SUNY*

**Penghang Yin**[†]                                                                                 *pyin@albany.edu*
*Department of Mathematics and Statistics*
*University at Albany, SUNY*

**Reviewed on OpenReview:** *https://openreview.net/forum?id=FHnTRAAdAZ*

## Abstract

Fine-tuning large language models (LLMs) using low-rank adaptation (LoRA) has become a highly efficient approach for downstream tasks, particularly in scenarios with limited computational resources. However, applying LoRA techniques to quantized LLMs poses unique challenges due to the reduced representational precision of quantized weights. In this paper, we introduce CLoQ (**C**alibrated **Lo**RA initialization for **Q**uantized LLMs), a simplistic initialization strategy designed to overcome these challenges. Our approach focuses on minimizing the layer-wise discrepancy between the original LLM and its quantized counterpart with LoRA components during initialization. By leveraging a small calibration dataset, CLoQ quantizes a pre-trained LLM and determines the optimal LoRA components for each layer, ensuring a strong foundation for subsequent fine-tuning. A key contribution of this work is a novel theoretical result that enables the accurate and closed-form construction of these optimal LoRA components. We validate the efficacy of CLoQ across multiple tasks such as language generation, arithmetic reasoning, and commonsense reasoning, demonstrating that it consistently outperforms existing LoRA fine-tuning methods for quantized LLMs, especially at 2-bit. The code is available at `https://github.com/AozhongZhang/CLoQ`

---

[*]Equal contribution.
[†]Corresponding to: `pyin@albany.edu`.

# 1 Introduction

Large language models (LLMs) Achiam et al. (2023); Touvron et al. (2023); Jiang et al. (2023); Guo et al. (2024a) have achieved remarkable success across a wide range of domains and applications. With ongoing advancements, the size and complexity of LLMs have grown exponentially, with some models now exceeding billions or even trillions of parameters. Although this scaling has unlocked unprecedented capabilities, it also introduces significant challenges, particularly in efficiently adapting these models to downstream tasks. Traditionally, full fine-tuning has been the dominant approach for adapting pre-trained models, involving updates to all model parameters. While effective in achieving state-of-the-art results, full fine-tuning is resource-intensive, requiring substantial GPU memory to store both model weights and optimizer states. These memory demands grow with the size of the model, making full fine-tuning increasingly impractical for large-scale models in resource-constrained settings.

To address these challenges, parameter-efficient fine-tuning (PEFT) Houlsby et al. (2019), such as Low-Rank Adaptation (LoRA) Hu et al. (2021) , has emerged as a promising approach. PEFT updates only a small subset of parameters while keeping the majority of the model unchanged, enabling resource-efficient fine-tuning of large-scale models. LoRA, for instance, introduces small, learnable low-rank matrices into the model's architecture. These matrices are optimized during fine-tuning while the original model weights remain frozen, significantly reducing memory and computational requirements. This design leverages the insight that weight updates often reside in a low dimensional subspace, allowing LoRA to achieve efficient adaptation with minimal overhead.

In an orthogonal direction, model compression techniques , notably quantization Rastegari et al. (2016); Hubara et al. (2018); Choi et al. (2018); Wang et al. (2018); Yin et al. (2016; 2018; 2019b;a); Li et al. (2023b); Shao et al. (2023); Zhang et al. (2023); Frantar et al. (2022a); Zhang et al. (2024a), have been developed to minimize GPU memory usage by converting high-precision weights into low-precision representations. Initially designed for inference in memory-limited environments, quantization methods have since been adapted to support fine-tuning. A notable advancement in this regard is QLoRA Dettmers et al. (2023) , which combines LoRA with quantization techniques to reduce GPU memory requirements for fine-tuning significantly. By leveraging low-rank adaptation and quantized weights, QLoRA enables resource-efficient fine-tuning of LLMs without compromising performance, making it a powerful tool for adapting large-scale models to downstream tasks. However, extending LoRA to quantized LLMs introduces additional challenges, as the reduced representational precision of quantized weights can disrupt standard initialization strategies, impacting task performance. Recent works Li et al. (2023a); Yao et al. (2023); Liao et al. (2024); Guo et al. (2024b) proposed minimizing quantization error through strategic initialization of the low-rank components in LoRA, to align with the original weight states of the model. This strategy has demonstrated success in fine-tuning lower-bit quantized LLMs.

**Contributions.** In this paper, we introduce CLoQ (**C**alibrated **LoRA** for **Q**uantized LLMs), an efficient layer-wise, data-driven initialization strategy specifically designed for quantized LLMs by leveraging a small calibration dataset. CLoQ consists of two main steps: a post-training quantization phase to obtain quantized weights and a generalized low-rank approximation phase under a linear transformation to compute the corresponding optimal adapters. We derive a novel closed-form solution to the low-rank approximation problem, which can be efficiently computed using just two singular value decompositions (SVDs).

Our CLoQ method requires no back-propagation, making it a highly efficient LoRA initialization scheme for fine-tuning quantized models. We demonstrate the effectiveness of CLoQ through extensive validation on multiple benchmark datasets. Our results show that CLoQ consistently outperforms existing LoRA methods for quantized LLMs, particularly at ultra-low bit-widths. For instance, the fine-tuning accuracy of `INT2` CLoQ on the Llama2-13B model Touvron et al. (2023) surpasses that of `INT4` QLoRA Dettmers et al. (2023) in the arithmetic reasoning tasks, as shown in Table 3.

**Notations.** We clarify the mathematical notations that will be used throughout this paper: we denote vectors by bold small letters and matrices by bold capital ones. For any matrix $\boldsymbol{X} \in \mathbb{R}^{m \times n}$, $\boldsymbol{X}^\top \in \mathbb{R}^{n \times m}$ is the transpose of $\boldsymbol{X}$, and $\text{Tr}(\boldsymbol{X}) := \sum_{i=1}^m X_{i,i}$ denotes the trace of $\boldsymbol{X}$ when $m = n$. For any two matrices $\boldsymbol{X}, \boldsymbol{Y} \in \mathbb{R}^{m \times n}$, $\langle \boldsymbol{X}, \boldsymbol{Y} \rangle := \text{Tr}(\boldsymbol{X}^\top \boldsymbol{Y}) = \sum_{i=1}^m \sum_{j=1}^n X_{i,j} Y_{i,j}$ is the inner product. We denote the

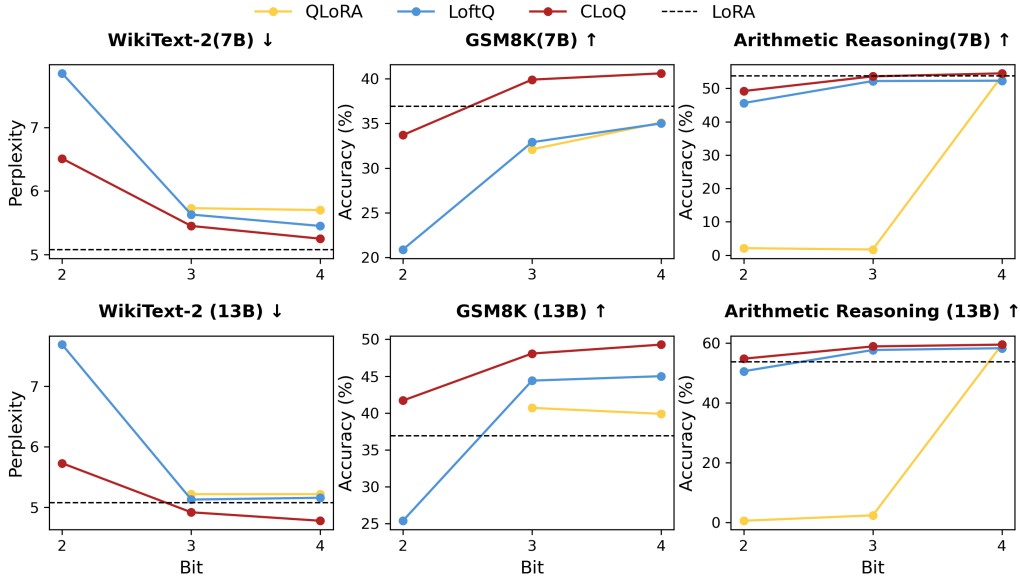

Figure 1: Fine-tuning results of Llama2-7B and Llama2-13B across various tasks. Left: the perplexity measured on WikiText-2. Middle: the accuracy achieved on GSM8K. Right: the average accuracy across multiple arithmetic reasoning tasks.

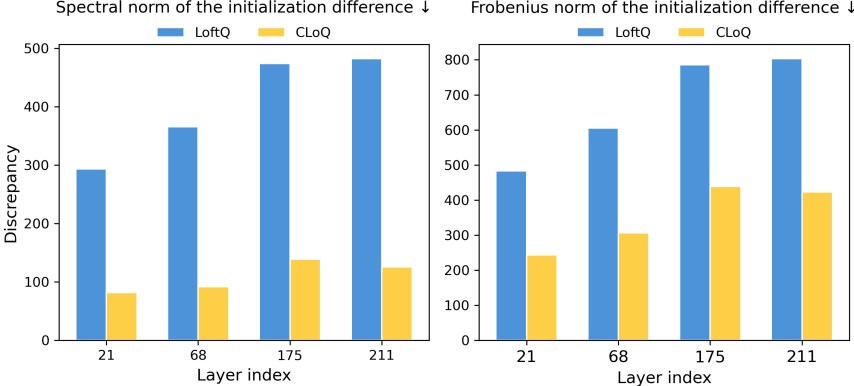

Figure 2: The discrepancy $\|\boldsymbol{X}(\boldsymbol{Q} + \boldsymbol{AB}^T - \boldsymbol{W})\|$ between the LoRA initialization and the original pre-trained weight matrix, computed using the spectral norm and the Frobenius norm, respectively. The layer shown in the figures above is randomly selected from the Llama2-7B model. The initialization is derived using both CLoQ and LoftQ under `INT2` quantization. Notably, CLoQ significantly reduces this discrepancy, demonstrating its effectiveness.

Frobenius norm of $\boldsymbol{X}$ by $\|\boldsymbol{X}\|_{\mathrm{F}} := \sqrt{\mathrm{Tr}(\boldsymbol{X}^{\top}\boldsymbol{X})} = \sqrt{\sum_{i=1}^{m}\sum_{j=1}^{n} X_{i,j}^2}$. In addition, for any diagonal matrix $\boldsymbol{\Sigma} = \mathrm{diag}(\sigma_1, \ldots, \sigma_n)$, we denote its square root by $\boldsymbol{\Sigma}^{\frac{1}{2}} := \mathrm{diag}(\sigma_1^{\frac{1}{2}}, \ldots, \sigma_n^{\frac{1}{2}})$.

## 2   Background

**Integer Quantizer.** Given a set of $m$ weights $\boldsymbol{w} \in \mathbb{R}^m$, the widely-used $b$-bit uniform integer (INT) quantizer Choi et al. (2018) determines the float scaling factor $\delta = \frac{\max(\boldsymbol{w}) - \min(\boldsymbol{w})}{2^b - 1}$ and zero-point $z = -\left\lfloor \frac{\min(\boldsymbol{w})}{\delta} \right\rceil$, where $\lfloor \cdot \rceil$ is nearest-to-round operation. The quantizer projects $\boldsymbol{w}$ onto the equally spaced grids $\mathbb{Q} = \{z \cdot \delta, (z+1) \cdot \delta, \ldots, (z + (2^b - 1)) \cdot \delta\}^m$ to obtain the quantized weights

$$\boldsymbol{q} = \delta \cdot \left( \mathrm{clip}\left( \left\lfloor \frac{\boldsymbol{w}}{\delta} \right\rceil + z, 0, 2^b - 1 \right) - z \right).$$

For channel-wise (or group-wise) quantization, the scaling factor $\delta$ is shared by the quantized weights within the same channel (or group, respectively).

**Post-Training Quantization.** Post-training quantization (PTQ) has been a cornerstone technique for compressing LLMs. PTQ methods directly identify low-precision representations of a model without requiring retraining, making them particularly well-suited for extremely large-scale AI models including LLMs. The simplest approach in this category is data-free quantization, commonly referred to as RTN, which involves rounding the pre-trained model weights to their nearest quantized states. More advanced PTQ algorithms, such as OPTQ Frantar et al. (2022b), leverage a small calibration dataset to solve a least-squares problem under discrete constraints:

$$\min_{\boldsymbol{Q} \in \mathbb{Q}} \|\boldsymbol{X}\boldsymbol{Q} - \boldsymbol{X}\boldsymbol{W}\|_{\mathrm{F}}^2, \tag{1}$$

to calibrate the quantization layer by layer, where $\boldsymbol{W}$ is the weight matrix of the pre-trained model. In this formulation, $\boldsymbol{X}$ denotes the activation or feature matrix corresponding to a batch of calibration data for a fixed layer. This approach ensures that the quantization process preserves the layer's output by minimizing the discrepancy between the original and quantized representations on the calibration dataset. Several efficient back-propagation-free algorithms Zhang et al. (2023); Behdin et al. (2023); Zhang et al. (2024b) have been proposed to address (1).

**Low-Rank Adaptation.** LoRA Hu et al. (2021) enables efficient fine-tuning of large pre-trained models by introducing two small, trainable matrices, $\boldsymbol{A}$ and $\boldsymbol{B}$, which are added to the frozen weight matrix $\boldsymbol{W}$. The weights of the fine-tuned model are then expressed as $\boldsymbol{W} + \boldsymbol{A}\boldsymbol{B}^\top$, where $\boldsymbol{A} \in \mathbb{R}^{m \times r}$, $\boldsymbol{B} \in \mathbb{R}^{n \times r}$, and $r \ll \min(m, n)$. During fine-tuning, only $\boldsymbol{A}$ and $\boldsymbol{B}$ are updated, while $\boldsymbol{W}$ remains fixed, significantly reducing the number of trainable parameters and computational overhead. LoRA initializes its parameters as follows:

$$\boldsymbol{A} \sim \mathcal{N}(0, \sigma^2), \quad \boldsymbol{B} = \boldsymbol{0},$$

ensuring that the initialization maintains perfect alignment with the pre-trained weights $\boldsymbol{W}$. Recent research has focused on developing variants of LoRA aimed at enhancing its performance Wang et al. (2024a;b); Liu et al. (2024); Wang et al. (2024c); Meng et al. (2024); Büyükakyüz (2024).

## 3   Method

CLoQ is designed to enhance the fine-tuning of quantized LLMs by incorporating calibration data, building upon OPTQ Frantar et al. (2022a) with a specific focus on the initialization of LoRA adapters. It utilizes second-order information derived from input activations $\boldsymbol{X}$ to ensure that the low-rank adapter matrices $\boldsymbol{A}$ and $\boldsymbol{B}$ are initialized in a manner that minimizes quantization error, particularly in relation to the data distribution. This alignment allows CLoQ to improve the fine-tuning process by making the low-rank adaptation more responsive to the model's behavior during calibration, thereby reducing the mismatch between the quantized model and its full-precision counterpart. Importantly, CLoQ uses the same calibration data as OPTQ, ensuring that the initialization of LoRA adapters is both universal and effective across various downstream tasks. In our experiments, we use the Wikitext dataset for calibration, offering a robust and generalizable foundation for fine-tuning across a range of task domains.

### 3.1 Calibrated Quantization and Low-Rank Initialization

Given the activation matrix $\boldsymbol{X}$ associated with the calibration data, CLoQ aims to solve the following optimization problem for LoRA initialization:

$$\min_{\boldsymbol{Q}\in\mathbb{Q},\boldsymbol{A}\in\mathbb{R}^{m\times r},\boldsymbol{B}\in\mathbb{R}^{n\times r}} \|\boldsymbol{X}(\boldsymbol{Q}+\boldsymbol{A}\boldsymbol{B}^\top-\boldsymbol{W})\|_\mathrm{F}^2, \tag{2}$$

where $\boldsymbol{X}\in\mathbb{R}^{(b\cdot l)\times m}$ is the activation matrix associated with a calibration data set of $b$ samples, each represented as an $l\times m$ sub-matrix and stacked along the row dimension. $\mathbb{Q}\subset\mathbb{R}^{m\times n}$ is an appropriate set of all feasible quantized weights. The objective of (2) is to ensure that the initialized weights for the LoRA model, $\boldsymbol{Q}+\boldsymbol{A}\boldsymbol{B}^\top$, closely approximate the pre-trained model weights $\boldsymbol{W}\in\mathbb{R}^{m\times n}$ when applied to the activation matrix.

This problem can, in theory, be solved using an alternating minimization (AltMin) method, whose $t$-th iteration reads:

$$\boldsymbol{Q}^{t+1} = \arg\min_{\boldsymbol{Q}\in\mathbb{Q}} \|\boldsymbol{X}(\boldsymbol{Q}+\boldsymbol{A}^t(\boldsymbol{B}^t)^\top-\boldsymbol{W})\|_\mathrm{F}^2$$
$$\boldsymbol{A}^{t+1},\boldsymbol{B}^{t+1} = \arg\min_{\boldsymbol{A},\boldsymbol{B}} \|\boldsymbol{X}(\boldsymbol{A}\boldsymbol{B}^\top+\boldsymbol{Q}^{t+1}-\boldsymbol{W})\|_\mathrm{F}^2$$

In practice, we observe that it suffices to just perform *a single iteration* with the initialization $\boldsymbol{A}^0(\boldsymbol{B}^0)^\top=0$. Therefore, the proposed CLoQ method comprises two steps: a quantization step and a low-rank approximation step under a linear transformation, which we detail in the rest of this section.

After solving the above problem and obtaining $\boldsymbol{Q}$, $\boldsymbol{A}$, $\boldsymbol{B}$, we fix the quantized weights $\boldsymbol{Q}$ and update only the low-rank components $\boldsymbol{A}$, $\boldsymbol{B}$ during the subsequent fine-tuning stage, as per the LoRA approach.

#### 3.1.1 Post-Training Quantization

With $\boldsymbol{A}\boldsymbol{B}^\top$ initialized to zero, the problem of finding $\boldsymbol{Q}$ simplifies to the standard layer-wise post-training quantization (PTQ):

$$\min_{\boldsymbol{Q}\in\mathbb{Q}} \|\boldsymbol{X}(\boldsymbol{Q}-\boldsymbol{W})\|_\mathrm{F}^2, \tag{3}$$

an optimization problem that has been extensively studied in recent literature Frantar et al. (2022b); Zhang et al. (2023); Chee et al. (2023); Xiao et al. (2023); Zhang et al. (2024b). For this, we adopt the widely-used OPTQ method Frantar et al. (2022a). To further enhance OPTQ's performance, we incorporate a preprocessing technique called weight magnitude reduction (MagR) Zhang et al. (2024a), which modifies $\boldsymbol{W}$ by removing outliers prior to quantization. MagR preprocessing significantly improves OPTQ's effectiveness in the low-bit regime while introducing minimal additional time beyond the OPTQ process and incurring no computational or memory overhead during inference time.

#### 3.1.2 Generalized Low-rank Approximation

After obtaining $\boldsymbol{Q}$ in the quantization step, we denote by

$$\Delta\boldsymbol{W} = \boldsymbol{W} - \boldsymbol{Q}$$

the residual of the quantized weights. To determine $\boldsymbol{A}$ and $\boldsymbol{B}$, we solve the following low-rank approximation problem under the linear transformation induced by $\boldsymbol{X}$:

$$\min_{\boldsymbol{A}\in\mathbb{R}^{m\times r},\boldsymbol{B}\in\mathbb{R}^{n\times r}} \|\boldsymbol{X}(\boldsymbol{A}\boldsymbol{B}^\top-\Delta\boldsymbol{W})\|_\mathrm{F}^2. \tag{4}$$

It is important to note that problem (4) is non-trivial due to the presence of the matrix $\boldsymbol{X}$, and its optimal solution is not given by directly computing the low-rank approximation (or SVD) of $\Delta\boldsymbol{W}$. However, we demonstrate in the following result that the problem can be *solved accurately by performing just two SVDs*:

**Theorem 3.1.** *Suppose the activation matrix $\boldsymbol{X} \in \mathbb{R}^{(b \cdot l) \times m}$ ($b \cdot l \gg m$) is of full-rank. Suppose the Gram (or Hessian) matrix $\boldsymbol{H} = \boldsymbol{X}^\top \boldsymbol{X} \in \mathbb{R}^{m \times m}$ has the SVD $\boldsymbol{H} = \boldsymbol{U}_H \boldsymbol{\Sigma}_H \boldsymbol{U}_H^\top$ and denote by $\boldsymbol{R} = \boldsymbol{\Sigma}_H^{\frac{1}{2}} \boldsymbol{U}_H^\top$ the non-symmetric root of $\boldsymbol{H}$. Then any pair $(\boldsymbol{A}, \boldsymbol{B})$ satisfying*

$$\boldsymbol{A}\boldsymbol{B}^\top = \boldsymbol{R}^{-1}\mathrm{LR}_r(\boldsymbol{R}\Delta\boldsymbol{W}). \tag{5}$$

*permits an optimal solution to problem (4). Here $\mathrm{LR}_r(\boldsymbol{R}\Delta\boldsymbol{W}) = \arg\min_{\mathrm{rank}(\boldsymbol{Z}) \leq r} \|\boldsymbol{Z} - \boldsymbol{R}\Delta\boldsymbol{W}\|_\mathrm{F}^2$ denotes the best rank-r approximation of $\boldsymbol{R}\Delta\boldsymbol{W}$.*

*Proof.* Firstly, we observe that the objective in (4) has the following equivalent expression:

$$
\begin{aligned}
&\|\boldsymbol{X}(\boldsymbol{A}\boldsymbol{B}^\top - \Delta\boldsymbol{W})\|_\mathrm{F}^2 \\
&= \mathrm{Tr}\big((\boldsymbol{A}\boldsymbol{B}^\top - \Delta\boldsymbol{W})^\top \boldsymbol{X}^\top \boldsymbol{X}(\boldsymbol{A}\boldsymbol{B}^\top - \Delta\boldsymbol{W})\big) \\
&= \mathrm{Tr}\big((\boldsymbol{A}\boldsymbol{B}^\top - \Delta\boldsymbol{W})^\top \boldsymbol{H}(\boldsymbol{A}\boldsymbol{B}^\top - \Delta\boldsymbol{W})\big) \\
&= \mathrm{Tr}\big((\boldsymbol{A}\boldsymbol{B}^\top - \Delta\boldsymbol{W})^\top \boldsymbol{R}^\top \boldsymbol{R}(\boldsymbol{A}\boldsymbol{B}^\top - \Delta\boldsymbol{W})\big) \\
&= \|\boldsymbol{R}(\boldsymbol{A}\boldsymbol{B}^\top - \Delta\boldsymbol{W})\|_\mathrm{F}^2 \\
&= \|\boldsymbol{R}\boldsymbol{A}\boldsymbol{B}^\top - \boldsymbol{R}\Delta\boldsymbol{W}\|_\mathrm{F}^2,
\end{aligned}
$$

where in the third equality, we used the identity: $\boldsymbol{H} = \boldsymbol{R}^\top \boldsymbol{R}$. Moreover, since $\boldsymbol{X}$ is full rank, $\boldsymbol{H}$ is invertible, and so is $\boldsymbol{R}$.

Based on these facts, we interpret problem (4) as finding the standard best rank-$r$ approximation of $\boldsymbol{R}\Delta\boldsymbol{W}$, $\mathrm{LR}_r(\boldsymbol{R}\Delta\boldsymbol{W})$, which can be obtained by performing the SVD of $\boldsymbol{R}\Delta\boldsymbol{W}$ and extracting the top-$r$ principal components Eckart & Young (1936). Then, $(\boldsymbol{A}, \boldsymbol{B})$ is an optimal solution to

$$\min_{\boldsymbol{A} \in \mathbb{R}^{m \times r}, \boldsymbol{B} \in \mathbb{R}^{n \times r}} \|\boldsymbol{R}\boldsymbol{A}\boldsymbol{B}^\top - \boldsymbol{R}\Delta\boldsymbol{W}\|_\mathrm{F}^2$$

if and only if

$$\boldsymbol{R}\boldsymbol{A}\boldsymbol{B}^\top = \mathrm{LR}_r(\boldsymbol{R}\Delta\boldsymbol{W}).$$

Consequently, since $\boldsymbol{R}$ is invertible, any $(\boldsymbol{A}, \boldsymbol{B})$ fulfilling

$$\boldsymbol{A}\boldsymbol{B}^\top = \boldsymbol{R}^{-1}\mathrm{LR}_r(\boldsymbol{R}\Delta\boldsymbol{W}).$$

permits an optimal solution to problem (4). $\qquad\square$

To apply Theorem 3.1 to solve problem (4), we make the following observations:

- One SVD is required to compute $\boldsymbol{R}$ and another is needed to determine $\mathrm{LR}_r(\boldsymbol{R}\Delta\boldsymbol{W})$. Given that $\boldsymbol{R} \in \mathbb{R}^{m \times m}$ and $\boldsymbol{R}\Delta\boldsymbol{W} \in \mathbb{R}^{m \times n}$, while $\boldsymbol{X} \in \mathbb{R}^{(b \cdot l) \times m}$, the computational complexity of SVDs required for solving (4) is independent of $b \cdot l$, which is significantly larger than $m$ or $n$ in practice. Here $l$ represents the context length, and $b$ denotes the size of the calibration dataset. We note that CLoQ requires fewer SVD computations than LoftQ Li et al. (2023a), which, by default, performs five AltMin iterations, each involving one SVD. Although CLoQ uses the more expensive GPTQ for the quantization (versus the data-free RTN in LoftQ), their overall initialization runtimes are comparable, as shown in Table 10.

- Indeed, (5) admits infinitely many optimal solutions. Suppose $\mathrm{LR}_r(\boldsymbol{R}\Delta\boldsymbol{W})$ has the form $\boldsymbol{U}_{:r}\boldsymbol{\Sigma}_{:r}\boldsymbol{V}_{:r}^\top$. In our experiments, we consistently take $\boldsymbol{A} = \boldsymbol{R}^{-1}\boldsymbol{U}_{:r}\boldsymbol{\Sigma}_{:r}$ and $\boldsymbol{B} = \boldsymbol{V}_{:r}$, which empirically performs well. However, it is clear that if $(\boldsymbol{A}, \boldsymbol{B})$ is an optimal solution, then for any invertible $\boldsymbol{C} \in \mathbb{R}^{r \times r}$, the pair $\big(\boldsymbol{A}\boldsymbol{C}, \boldsymbol{B}(\boldsymbol{C}^{-1})^\top\big)$ also satisfies (5) and thus provides an optimal solution to (4). For example, $(\boldsymbol{A}, \boldsymbol{B}) = (\boldsymbol{R}^{-1}\boldsymbol{U}_{:r}, \boldsymbol{V}_{:r}\boldsymbol{\Sigma}_{:r})$ or $(\boldsymbol{R}^{-1}\boldsymbol{U}_{:r}\boldsymbol{\Sigma}_{:r}^{\frac{1}{2}}, \boldsymbol{V}_{:r}\boldsymbol{\Sigma}_{:r}^{\frac{1}{2}})$. In Section 5, our ablation study shows that the combination $(\boldsymbol{R}^{-1}\boldsymbol{U}_{:r}\boldsymbol{\Sigma}_{:r}, \boldsymbol{V}_{:r})$ gives the best practical performance during the subsequent LoRA fine-tuning process. A more comprehensive theoretical analysis of the impact of initialization like Hayou et al. (2024); Li et al. (2024) will be addressed in future work.

- When $\boldsymbol{H}$ is not invertible or poorly conditioned, we propose adding a small constant $\lambda$ to the diagonal elements. Specifically, we typically set $\lambda = 0.01 \frac{\text{Tr}(\boldsymbol{H})}{m}$, following a strategy similar to that used in the prior works Frantar et al. (2022b); Chee et al. (2023). This adjustment has consistently proven effective in mitigating numerical issues and ensuring stability during computations.

- In theory, even if $\boldsymbol{X}$ is rank-deficient, the optimality condition $\boldsymbol{RAB}^\top = \text{LR}_r(\boldsymbol{R}\Delta\boldsymbol{W})$ in the proof of Theorem 3.1 still holds. But in this case, since $\boldsymbol{R}$ is also rank-deficient, this optimality condition permits infinitely many solution for $\boldsymbol{AB}^\top$, and we may simply take $\boldsymbol{AB}^\top = \boldsymbol{R}^\dagger \text{LR}_r(\boldsymbol{R}\Delta\boldsymbol{W})$, where $\boldsymbol{R}^\dagger$ is a pseudo-inverse.

To summarize, our proposed CLoQ method is detailed in Algorithm 1.

---

**Algorithm 1** CLoQ for initializing one linear layer

---

**Input:** Regularized Gram matrix of activations $\boldsymbol{H} = \boldsymbol{X}^\top\boldsymbol{X} + \lambda\boldsymbol{I} \in \mathbb{R}^{m \times m}$, Pre-trained weight matrix $\boldsymbol{W} \in \mathbb{R}^{m \times n}$, Rank $r \ll \min(m, n)$.
**Output:** Quantized weight matrix $\boldsymbol{Q} \in \mathbb{R}^{m \times n}$, Low-rank components $\boldsymbol{A} \in \mathbb{R}^{m \times r}$, $\boldsymbol{B} \in \mathbb{R}^{n \times r}$.

1: Solve (3) to obtain the quantized weights $\boldsymbol{Q}$.
2: Compute the residual of quantized weights $\Delta\boldsymbol{W} = \boldsymbol{W} - \boldsymbol{Q} \in \mathbb{R}^{m \times n}$
3: Perform SVD: $\boldsymbol{H} = \boldsymbol{U}_H\boldsymbol{\Sigma}_H\boldsymbol{U}_H^\top$
4: Evaluate: $\boldsymbol{R} = \boldsymbol{\Sigma}_H^{\frac{1}{2}}\boldsymbol{U}_H^\top$
5: Perform the SVD of $\boldsymbol{R}\Delta\boldsymbol{W}$ to find its best rank-$r$ approximation: $\text{LR}_r(\boldsymbol{R}\Delta\boldsymbol{W}) = \boldsymbol{U}_{:r}\boldsymbol{\Sigma}_{:r}\boldsymbol{V}_{:r}^\top$
6: Compute the low-rank components:

$$\boldsymbol{A} = \boldsymbol{R}^{-1}\boldsymbol{U}_{:r}\boldsymbol{\Sigma}_{:r}$$

$$\boldsymbol{B} = \boldsymbol{V}_{:r}$$

**Return:** $\boldsymbol{Q}, \boldsymbol{A}, \boldsymbol{B}$

---

## 4 Experiment

In this section, we evaluate the effectiveness of CLoQ on language modeling, arithmetic reasoning, and commonsense reasoning tasks. The fine-tuning through CLoQ consists of two key stages: the initialization step and the fine-tuning step. In the initialization step, we quantize the full-precision weight $\boldsymbol{W}$ into low-precision weight $\boldsymbol{Q}$ and find optimal low-rank matrices $\boldsymbol{A}$ and $\boldsymbol{B}$ that minimize the residual error. In the finetuning step, the quantized weight matrix $\boldsymbol{Q}$ is fixed in low-precision, while $\boldsymbol{A}$ and $\boldsymbol{B}$ are trained through back-propagation. The detailed hyperparameter settings for all our experiments are presented in the Appendix A.

**Models and Datasets.** We test CLoQ on Llama2-7b, Llama2-13b Touvron et al. (2023), Llama3-8b Grattafiori et al. (2024) and Mistral-7b-v0.1 Jiang et al. (2023) models. Following prior works Frantar et al. (2022a), we randomly sample 128 instances, each with a context length of 2048 tokens, from the WikiText-2 dataset Merity et al. (2016) to serve as the calibration set for quantization. Then, we fine-tune and evaluate the models on WikiText-2 for language modeling. For single arithmetic reasoning tasks, we fine-tune and evaluate on the GSM8K Cobbe et al. (2021). For multi arithmetic reasoning, we fine-tune the models on Math10K Hu et al. (2023) and then evaluate the test sets of AQuA Ling et al. (2017), GSM8K, MAWPS Koncel-Kedziorski et al. (2016) and SVAMP Patel et al. (2021). For commonsense reasoning tasks, we fine-tune the models on Commonsense170K Hu et al. (2023) and evaluate on eight representative tasks: BoolQ Clark et al. (2019), PIQA Bisk et al. (2020), SIQA Sap et al. (2019), HellaSwag Zellers et al. (2019), WinoGrande Sakaguchi et al. (2021), ARC-e, ARC-c Clark et al. (2018) and OBQA Mihaylov et al. (2018).

**Baselines.** We compare with LoRA Hu et al. (2021), QLoRA Dettmers et al. (2023), GPTQ-LoRA GPT (2023), LoftQ Li et al. (2023a) and ApiQ Liao et al. (2024). LoRA is often considered as the benchmark for fine-tuning performance. QLoRA incorporates NF-quantization, with its low-rank initialization aligning with the standard LoRA method. GPTQ-LoRA integrates OPTQ for the base weights and fine-tunes the LoRA component while preserving its low-rank initialization as in the standard LoRA, with the quantized

weights kept frozen. In contrast, LoftQ and LQ-LoRA carefully initialize the quantized weight and low-rank matrices by solving some optimization problems to minimize the approximation error. Furthermore, ApiQ uses gradient-based block-wise optimization to specifically initialize the low-rank components during post-training quantization.

### 4.1 Implementation details

**Quantization.** We quantize the weights of all linear layers in the base model using MagR preprocessing Zhang et al. (2024a) followed by OPTQ Frantar et al. (2022a). *The quantization scheme employs uniform (a.k.a. `INT`) and asymmetric quantization, with a default group size of 64.* After quantization, we compute the LoRA components $A$ and $B$, which are maintained in `FP16` precision.

**Fine-tuning.** Following prior works Dettmers et al. (2023); Li et al. (2023a); Liao et al. (2024), we fine-tune the models using the standard LoRA configuration, with modifications to the LoRA initialization and learning rates. The quantized weights remain fixed, and only the LoRA adapter matrices are trainable during fine-tuning. The rank of the LoRA adapters is consistently set to 64 across all methods. For optimization, we use AdamW Loshchilov (2017). All experiments are conducted on NVIDIA A100 GPUs with 80GB of memory.

### 4.2 Fine-tuning Results

**Language modeling.** We evaluate the models by reporting the perplexity of language generation on WikiText-2. As shown in Table 1 and 2, CLoQ consistently outperforms existing methods across most bit levels and model architectures. Notably, at `INT2`, CLoQ proves effective, achieving a perplexity reduction of 0.95 over ApiQ-lw and 1.34 over LoftQ on Llama2-7B, and even surpasses ApiQ-bw by 0.1 perplexity. This result highlights CLoQ's ability to maintain superior performance even under ultra-low bit quantization constraints.

Table 1: Finetuning results of WikiText and GSM8K on Llama2-7B and Llama2-13B.

| Method | Bit | Llama2-7B | | Llama2-13B | |
| | | Wiki (ppl↓) | GSM8K (acc↑) | Wiki (ppl↓) | GSM8K (acc↑) |
|---|---|---|---|---|---|
| LoRA | 16 | 5.08 | 36.9 | 5.12 | 45.3 |
| QLoRA | 4 | 5.70 | 35.1 | 5.22 | 39.9 |
| LoftQ | 4 | **5.24** | 35.0 | 5.16 | 45.0 |
| ApiQ-lw | 4 | 5.28 | 36.4 | 4.78 | 50.4 |
| ApiQ-bw | 4 | 5.27 | 39.8 | 4.78 | **51.2** |
| CLoQ | 4 | 5.25 | **40.6** | **4.78** | 49.3 |
| QLoRA | 3 | 5.73 | 32.1 | 5.22 | 40.7 |
| LoftQ | 3 | 5.63 | 32.9 | 5.13 | 44.4 |
| ApiQ-lw | 3 | 5.53 | 36.0 | 4.98 | 45.4 |
| ApiQ-bw | 3 | 5.49 | 39.3 | 4.96 | 47.6 |
| CLoQ | 3 | **5.45** | **39.9** | **4.92** | **48.1** |
| QLoRA | 2 | N.A. | N.A. | N.A. | N.A. |
| LoftQ | 2 | 7.85 | 20.9 | 7.69 | 25.4 |
| ApiQ-lw | 2 | 7.46 | 26.0 | 6.29 | 36.3 |
| ApiQ-bw | 2 | 6.61 | 33.5 | 5.79 | 41.2 |
| CLoQ | 2 | **6.51** | **33.7** | **5.73** | **41.7** |

Table 2: Finetuning results of WikiText and GSM8K on Llama3-8B and Mistral-7B.

| Method | Bit | Llama3-8B | | Mistral-7B | |
| | | Wiki (ppl↓) | GSM8K (acc↑) | Wiki (ppl↓) | GSM8K (acc↑) |
|---|---|---|---|---|---|
| LoRA | 16 | 6.34 | 47.8 | 5.17 | 52.8 |
| LoftQ | 2 | 14.09 | 31.6 | 1849.33 | 1.7 |
| ApiQ-lw | 2 | - | - | 7.18 | 41.3 |
| ApiQ-bw | 2 | 9.89 | 44.4 | 6.69 | **45.0** |
| CLoQ | 2 | **9.49** | **45.4** | **6.68** | **45.0** |

**Arithmetic reasoning (single task).** To evaluate the models on GSM8K, we extract numerical answers from the generated solutions and determine accuracy by analyzing these extracted values. As shown in Table 1 and 2, CLoQ achieves better performance across different model types and quantization bit levels. At `INT2`,

CLoQ reaches an accuracy of 33.7% on Llama2-7B, which CLoQ achieves a 7.7% improvement in performance compared with ApiQ-lw. Moreover, on the Llama3-8B model, CLoQ even outperforms ApiQ-bw by 1.0% at `INT2`. More remarkably, CLoQ achieves comprehensive superiority over all existing methods at `INT2` and `INT3` on Llama2-13B. On the Mistral-7B model, the accuracy of `INT2` CLoQ is better than ApiQ-lw and is comparable with ApiQ-bw.

**Arithmetic reasoning.** To evaluate CLoQ's effectiveness across multiple arithmetic reasoning tasks, we fine-tuned models on the Math10K dataset and assessed their performance on four separate math reasoning benchmarks, demonstrating their adaptability to diverse mathematical challenges. As shown in Table 3 and 4, CLoQ consistently outperforms other methods across various model architectures and quantization levels, achieving superior accuracy both on average and on individual datasets. Notably, at `INT2` quantization, CLoQ delivers substantial gains, surpassing ApiQ-bw by 1.9% on LLaMA2-7B and by 2.1% on both LLaMA3-8B and LLaMA2-13B. Even at `INT4`, CLoQ surpasses both LoRA and QLoRA on Llama2-7B. Moreover, for complex reasoning tasks such as GSM8K and SVAMP, CLoQ achieves 4.1% and 1.7% higher accuracy than ApiQ-bw at `INT2` on Llama3-8B, highlighting its robustness in challenging problem settings. These results underscore CLoQ's remarkable potential in handling intricate reasoning tasks with enhanced accuracy, even under ultra-low-bit quantization.

Table 3: Accuracy on four arithmetic reasoning tasks. The LoRA rank $r$ is 64 for all methods.

| Method | Bit | Llama2-7B | | | | | Llama2-13B | | | | |
|---|---|---|---|---|---|---|---|---|---|---|---|
| | | GSM8K | SVAMP | MAWPS | AQuA | Avg. ↑ | GSM8K | SVAMP | MAWPS | AQuA | Avg. ↑ |
| LoRA | 16 | 43.6 | 59.4 | 85.0 | 27.0 | 53.7 | 55.3 | 67.7 | 87.4 | 24.4 | 58.7 |
| QLoRA | 4 | 42.7 | 58.7 | **87.3** | 26.4 | 53.7 | 54.8 | **69.4** | 87.0 | 26.8 | **59.5** |
| GPTQ-LoRA | 4 | 43.0 | 58.4 | 86.1 | 24.3 | 52.9 | 53.2 | 67.5 | 85.3 | 25.6 | 57.9 |
| LoftQ | 4 | 41.7 | 56.0 | 86.3 | 25.3 | 52.3 | 54.9 | 66.5 | 87.7 | 23.9 | 58.3 |
| ApiQ-bw | 4 | 43.2 | 59.0 | 85.7 | 26.0 | 53.5 | 55.3 | 67.4 | 87.8 | 25.6 | 59.0 |
| CLoQ | 4 | **43.6** | **60.3** | 87.0 | **27.6** | **54.6** | **55.4** | 66.9 | **88.7** | **27.2** | **59.5** |
| QLoRA | 3 | 1.4 | 1.4 | 0.7 | 3.4 | 1.7 | 0.8 | 2.5 | 0.3 | 6.2 | 2.4 |
| GPTQ-LoRA | 3 | 38.9 | 55.7 | 84.9 | 23.2 | 50.7 | 50.6 | 65.2 | 88.0 | 22.6 | 56.6 |
| LoftQ | 3 | 39.9 | 56.3 | 86.3 | **26.4** | 52.2 | **53.9** | 66.1 | 87.0 | 23.6 | 57.7 |
| ApiQ-bw | 3 | 41.4 | 55.9 | 87.0 | 25.2 | 52.4 | 51.5 | **67.4** | 88.5 | 25.6 | 58.3 |
| CLoQ | 3 | **42.2** | **58.9** | **89.1** | 24.0 | **53.6** | 52.5 | 66.1 | **89.1** | **28.0** | **58.9** |
| QLoRA | 2 | 0.9 | 1.5 | 0.8 | 5.1 | 2.1 | 0.5 | 0.7 | 0.1 | 0.9 | 0.6 |
| GPTQ-LoRA | 2 | 21.7 | 39.0 | 76.6 | 22.1 | 39.9 | 31.9 | 49.6 | 82.5 | 23.6 | 46.9 |
| LoftQ | 2 | 29.5 | 45.8 | 83.6 | 23.2 | 45.6 | 37.0 | 55.9 | 87.7 | 21.7 | 50.6 |
| ApiQ-bw | 2 | 31.2 | 51.0 | 82.9 | 23.9 | 47.3 | 43.1 | **59.2** | 85.1 | 23.4 | 52.7 |
| CLoQ | 2 | **34.7** | **52.0** | **86.1** | **24.1** | **49.2** | **44.6** | 57.6 | **88.7** | **28.4** | **54.8** |

Table 4: Accuracy on four arithmetic reasoning tasks. The LoRA rank $r$ is 64 for all methods. CLoQ accuracies are averaged over 5 random runs, with standard deviations reported.

| Method | Bit | Llama3-8B | | | | |
|---|---|---|---|---|---|---|
| | | GSM8K | SVAMP | MAWPS | AQuA | Avg. ↑ |
| LoRA | 16 | 68.8 | 76.1 | 90.3 | 31.5 | 66.7 |
| LoftQ | 2 | 35.6 | 52.1 | 87.0 | 25.2 | 50.0 |
| ApiQ-bw | 2 | 47.0 | 67.2 | 88.2 | 27.2 | 57.4 |
| CLoQ | 2 | $50.7_{\pm 0.35}$ | $67.5_{\pm 0.86}$ | $88.3_{\pm 0.91}$ | $29.5_{\pm 0.56}$ | $59.0_{\pm 0.34}$ |

**Commonsense reasoning.** We further evaluate the effectiveness of CLoQ on commonsense reasoning tasks by fine-tuning the models on the Commonsense170K dataset and testing their performance across eight reasoning benchmarks, as presented in Table 5. Consistent with its performance on arithmetic reasoning tasks, CLoQ outperforms other methods across different model sizes and quantization levels, achieving notable improvements both on average and within each dataset.

At `INT2`, CLoQ delivers a performance boost over ApiQ-bw, with an average accuracy improvement exceeding 0.7%, even approaching the performance levels typically observed with `INT4` QLoRA. For instance, `INT2` CLoQ exhibits only a minimal average accuracy drop of 0.04% compared to `INT4` QLoRA. Additionally, at `INT4`, CLoQ attains impressive accuracy, reducing the gap to `FP16` LoRA to just 0.4%. These results indicate

that CLoQ significantly enhances LoRA's learning capacity, enabling it to better adapt to a diverse range of tasks.

Table 5: Accuracy on eight commonsense reasoning tasks. The LoRA rank $r = 64$ for all methods.

| Model | Method | Bit | BoolQ | PIQA | SIQA | HellaS. | WinoG. | ARC-e | ARC-c | OBQA | Avg. ↑ |
|---|---|---|---|---|---|---|---|---|---|---|---|
| | LoRA | 16 | 73.6 | 86.5 | 81.8 | 95.2 | 86.9 | 89.4 | 76.7 | 86.7 | 84.6 |
| | QLoRA | 4 | 73.9 | 84.4 | 79.7 | 93.3 | 84.6 | 86.1 | 73.0 | 85.1 | 82.5 |
| | GPTQ-LoRA | 4 | 73.4 | 83.6 | 79.3 | 93.3 | 84.5 | 86.5 | 72.8 | 83.3 | 82.1 |
| | LoftQ | 4 | 73.7 | 86.0 | 81.1 | 94.6 | 86.3 | 88.1 | 75.5 | 86.2 | 83.9 |
| | ApiQ-bw | 4 | 73.5 | **87.0** | **82.0** | **95.2** | **86.9** | **89.5** | **77.0** | 86.2 | **84.7** |
| | CLoQ | 4 | **74.2** | 86.3 | 81.6 | 95.1 | 85.9 | 88.7 | 75.7 | **86.4** | 84.2 |
| Llama2-7B | GPTQ-LoRA | 3 | 71.8 | 82.7 | 79.3 | 92.1 | 82.8 | 84.2 | 70.6 | 83.4 | 80.8 |
| | LoftQ | 3 | **74.0** | 85.6 | 81.0 | 94.3 | 85.6 | 88.1 | **75.4** | 85.5 | 83.7 |
| | ApiQ-bw | 3 | 73.3 | 85.6 | **81.8** | 94.6 | **86.9** | 87.9 | 73.7 | **86.4** | **83.8** |
| | CLoQ | 3 | 73.5 | **86.1** | 81.2 | **94.8** | 85.4 | **88.6** | 75.1 | 85.0 | 83.7 |
| | GPTQ-LoRA | 2 | 62.2 | 49.5 | 33.3 | 25.1 | 49.4 | 25.0 | 22.6 | 27.6 | 36.8 |
| | LoftQ | 2 | 62.4 | 70.5 | 73.4 | 78.8 | 71.0 | 66.5 | 50.8 | 62.3 | 67.0 |
| | ApiQ-bw | 2 | 68.4 | 80.7 | **79.6** | 91.4 | **82.4** | 82.7 | 68.3 | 80.5 | 79.3 |
| | CLoQ | 2 | **70.2** | **82.2** | 78.9 | **91.7** | 82.2 | **83.6** | **70.7** | **81.4** | **80.1** |
| | LoRA | 16 | 76.3 | 88.5 | 83.4 | 96.5 | 89.6 | 92.8 | 81.7 | 89.6 | 87.3 |
| | QLoRA | 4 | 74.9 | 86.6 | 81.5 | 94.9 | 86.9 | 89.1 | 77.1 | 87.2 | 84.8 |
| | GPTQ-LoRA | 4 | 74.5 | 86.1 | 81.8 | 94.7 | 86.8 | 89.0 | 77.1 | 84.5 | 84.3 |
| | LoftQ | 4 | 76.0 | 87.9 | 82.8 | 95.8 | 88.9 | 91.2 | 80.8 | 88.8 | 86.5 |
| | ApiQ-bw | 4 | **76.2** | **88.5** | **83.5** | **96.6** | **90.0** | **92.1** | 81.2 | 89.9 | **87.3** |
| | CLoQ | 4 | 75.7 | 88.4 | 82.9 | 96.3 | 89.4 | 91.1 | **81.9** | **90.0** | 87.0 |
| Llama2-13B | GPTQ-LoRA | 3 | 73.5 | 85.2 | 81.1 | 94.1 | 85.7 | 87.9 | 75.5 | 85.3 | 83.5 |
| | LoftQ | 3 | 75.2 | 87.8 | 82.8 | **96.3** | 89.5 | 91.1 | **81.4** | 88.0 | 86.5 |
| | ApiQ-bw | 3 | **76.0** | 88.0 | 82.3 | 95.8 | 89.1 | 91.1 | 81.1 | 89.5 | 86.6 |
| | CLoQ | 3 | 75.3 | **88.1** | 82.8 | 95.9 | **90.1** | **91.2** | 80.7 | **90.6** | **86.8** |
| | GPTQ-LoRA | 2 | 62.2 | 50.1 | 34.0 | 25.1 | 49.6 | 25.0 | 22.7 | 27.6 | 37.1 |
| | LoftQ | 2 | 65.9 | 76.4 | 78.0 | 84.4 | 76.1 | 75.1 | 60.1 | 72.7 | 73.6 |
| | ApiQ-bw | 2 | 73.1 | 85.2 | **82.3** | 94.4 | 86.2 | 88.2 | 74.9 | **85.9** | 83.8 |
| | CLoQ | 2 | **73.9** | **85.5** | 81.6 | **94.8** | **87.3** | **89.5** | **77.4** | 84.8 | **84.4** |

**Fine-tuning on mixed dataset.** We evaluate the performance of CLoQ for LoRA fine-tuning on a mixed dataset comprising Math10K and 5K samples from a Commonsense dataset, and assess accuracy on four arithmetic reasoning tasks. Interestingly, we find that fine-tuning on the mixed dataset leads to a drop in arithmetic reasoning accuracy compared to fine-tuning solely on the Math10K dataset, as shown by Table 6. For example, for 4-bit Llama2-7B, the averaged accuracy of CLoQ fine-tuned on mixed dataset drops to 51.2% from 54.6% (in Table 3). However, CLoQ still consistently outperforms LoftQ in this setting across different bit-widths.

Table 6: Accuracy of Llama2-7B for four arithmetic reasoning finetuned on mixed dataset.

| Method | Bit | Arithmetic reasoning | | | | |
|---|---|---|---|---|---|---|
| | | GSM8K | SVAMP | MAWPS | AQuA | Avg. ↑ |
| LoftQ | 4 | 36.8 | **55.3** | 83.2 | 24.8 | 50.0 |
| CLoQ | 4 | **38.8** | 54.9 | **85.7** | **25.2** | **51.2** |
| LoftQ | 2 | 21.8 | 40.7 | 74.4 | 24.0 | 40.2 |
| CLoQ | 2 | **27.2** | **47.0** | **80.3** | **24.4** | **44.7** |

## 4.3 Ablation study

**LoRA initialization with different $(\boldsymbol{A}, \boldsymbol{B})$ combinations.** Furthermore, we investigate the performance of various combinations of $(\boldsymbol{A}, \boldsymbol{B})$ in Algorithm 1, as shown in Table 7. We tried three different combinations of $(\boldsymbol{A}, \boldsymbol{B})$, including $(\boldsymbol{R}^{-1}\boldsymbol{U}_{:r}, \boldsymbol{V}_{:r}\boldsymbol{\Sigma}_{:r})$, $(\boldsymbol{R}^{-1}\boldsymbol{U}_{:r}\boldsymbol{\Sigma}_{:r}^{\frac{1}{2}}, \boldsymbol{V}_{:r}\boldsymbol{\Sigma}_{:r}^{\frac{1}{2}})$, and the default choice $(\boldsymbol{R}^{-1}\boldsymbol{U}_{:r}\boldsymbol{\Sigma}_{:r}, \boldsymbol{V}_{:r})$. Table 7 shows that the default combination of initialized adapters gives the best performance in the subsequent fine-tuning phase.

**Calibration data size.** We explore the sensitivity of CLoQ to the size of the calibration dataset. As shown in Table 8, CLoQ exhibits strong robustness to the calibration size. Across both INT4 and INT2, the overall

Table 7: Fine-tuning results of different combinations of $(\boldsymbol{A}, \boldsymbol{B})$ on WikiText-2 and GSM8K. The LoRA rank is $r = 64$.

| | | Llama2-7B | |
|---|---|---|---|
| Method | Bit | Wiki (ppl↓) | GSM8K (acc↑) |
| LoRA | 16 | 5.08 | 36.9 |
| $(\boldsymbol{R}^{-1}\boldsymbol{U}_{:r}, \boldsymbol{V}_{:r}\boldsymbol{\Sigma}_{:r})$ | 2 | 880.6 | 1.6 |
| $(\boldsymbol{R}^{-1}\boldsymbol{U}_{:r}\boldsymbol{\Sigma}_{:r}^{\frac{1}{2}}, \boldsymbol{V}_{:r}\boldsymbol{\Sigma}_{:r}^{\frac{1}{2}})$ | 2 | 6.68 | 12.9 |
| $(\boldsymbol{R}^{-1}\boldsymbol{U}_{:r}\boldsymbol{\Sigma}_{:r}, \boldsymbol{V}_{:r})$ | 2 | 6.51 | 33.7 |

performance remains consistently stable as the calibration size varies from 32 to 256. A calibration dataset of 128 samples, commonly used as the default in PTQ, yields slightly better results. These findings indicate that CLoQ does not heavily depend on the specific choice of calibration set size, which enhances its practicality and ease of deployment in real-world scenarios.

Table 8: Accuracy of different calibration dataset sizes for Llama2-7B.

| | | | | Arithmetic reasoning | | | | |
|---|---|---|---|---|---|---|---|---|
| Calibration Size | Bit | Wiki. ↓ | GSM8K ↑ | GSM8K | SVAMP | MAWPS | AQuA | Avg. ↑ |
| 32 | 4 | 5.34 | 40.2 | 43.4 | 58.7 | 87.4 | 28.0 | 54.4 |
| 64 | 4 | 5.33 | **40.9** | 44.7 | 57.6 | 87.0 | 25.2 | 53.6 |
| 128 (default) | 4 | **5.25** | 40.6 | 43.6 | 60.3 | 87.0 | 27.6 | **54.6** |
| 256 | 4 | 5.33 | 40.1 | 42.6 | 59.6 | 87.0 | 26.0 | 53.8 |
| 32 | 2 | 6.62 | 32.5 | 35.5 | 50.5 | 86.6 | 24.0 | 49.1 |
| 64 | 2 | 6.56 | 33.5 | 35.5 | 54.0 | 84.5 | 25.6 | **49.9** |
| 128 | 2 | 6.51 | **33.7** | 34.7 | 52.0 | 86.1 | 24.1 | 49.2 |
| 256 | 2 | **6.49** | 33.2 | 33.5 | 47.2 | 87.4 | 27.6 | 48.9 |

**Sequence length.** Additionally, we present the results for different sequence lengths when finetuning the 2-bit Llama2-7B model in Table 9 on arithmetic reasoning tasks. It shows that the fine-tuning accuracy nicely improves as the sequence length increases.

Table 9: Accuracy of finetuned 2-bit Llama2-7B for four arithmetic reasoning with different sequence lengths.

| | Arithmetic reasoning | | | | |
|---|---|---|---|---|---|
| Sequence length | GSM8K | SVAMP | MAWPS | AQuA | Avg. ↑ |
| 256 | 35.0 | 50.7 | 87.0 | 22.1 | 48.7 |
| 512 | 34.7 | 52.0 | 86.1 | 24.1 | 49.2 |
| 1024 | 34.1 | 52.8 | 87.8 | 24.0 | 49.7 |
| 2048 | 34.5 | 52.9 | 87.8 | 24.6 | 50.0 |

**Initialization Cost and Latency/Memory Benefits.** We compare the duration and GPU memory usage of Initialization between CLoQ and other baseline methods. As shown in Table 10, CLoQ achieves efficient and scalable initialization, offering notable reductions in both runtime and memory consumption. For instance, CLoQ requires only 0.7 hours and 9GB of memory on LLaMA2-7B, outperforming ApiQ-lw (4.1h) and ApiQ-bw (1.3h) by large margins. On the larger LLaMA2-13B, CLoQ remains competitive, requiring just 1.5 hours and 13GB of memory, compared to 6.5 hours (ApiQ-lw) and 27GB (LoftQ). These results demonstrates CLoQ's strong practicality for large-scale model under limited hardware budgets.

## 5    Related Work

When quantizing pre-trained models, QLoRA Dettmers et al. (2023) primarily emphasizes the quantization process, often overlooking the critical importance of subsequent LoRA fine-tuning. It adopts the fixup initialization strategy used in LoRA, attaching zero-initialized low-rank adapters to the quantized pre-trained model. However, the discrepancies introduced by quantization, especially in extremely low-bit regimes, can significantly impact the initialization of LoRA fine-tuning, ultimately affecting the overall fine-tuning

Table 10: The duration and peak GPU memory used for Llama2.

| Size | Method | Duration | Peak GPU memory |
|---|---|---|---|
| Llama2-7B | LoftQ | 0.6h | 14GB |
| | ApiQ-lw | 4.1h | 6GB |
| | ApiQ-bw | 1.3h | 12GB |
| | CLoQ | 0.7h | 9GB |
| Llama2-13B | LoftQ | 1.3h | 27GB |
| | ApiQ-lw | 6.5h | 9GB |
| | ApiQ-bw | 2.4h | 17GB |
| | CLoQ | 1.5h | 13GB |

performance. LoftQ Li et al. (2023a) jointly optimizes the quantized weights $Q$ and the low-rank adapter matrices $A$ and $B$ by solving the following optimization problem:

$$\min_{Q,A,B} \|Q + AB^\top - W\|_F^2. \tag{6}$$

This approach ensures that $Q$, $A$, and $B$ are initialized to minimize the reconstruction error between the quantized and pre-trained weights. By aligning the quantized model's initial state more closely with its pre-trained counterpart, LoftQ enhances fine-tuning performance without requiring calibration data.

LQ-LoRA Guo et al. (2024b) assigns importance weights to each parameter by evaluating its sensitivity to output variations, thereby guiding the decomposition process toward critical regions. Specifically, the (diagonal) Fisher matrix is used to weight the reconstruction objective during decomposition. To solve the resulting weighted SVD problem, LQ-LoRA assumes row and column homogeneity in the Fisher matrix, allowing the use of standard SVD techniques at the cost of theoretical precision. However, this approach has limitations. It relies on approximations rather than solving the weighted SVD problem exactly, which may lead to suboptimal results. Furthermore, computing the Fisher matrix requires back-propagation through the pre-trained model, introducing additional computational overhead. Similar to our work, ApiQ Liao et al. (2024) also employs an activation-aware initialization strategy utilizing calibration data. However, it relies on two activation matrices—one obtained from the pre-trained model and the other from the quantized model, whereas CLoQ uses only a single pre-trained activation matrix. ApiQ optimizes the discrepancy using standard back-propagation, whereas CLoQ adopts a fully gradient-free approach. By avoiding the computational overhead of back-propagation, CLoQ enables faster adaptation while maintaining high performance across various tasks.

# 6 Concluding Remarks

In this work, we introduced CLoQ, an efficient and scalable method for fine-tuning quantized LLMs. By leveraging a small calibration dataset, CLoQ optimally initializes LoRA adapters through a novel layer-wise, data-driven approach, significantly improving the fine-tuning process without the need for back-propagation. The use of a closed-form solution for low-rank approximation, computed via two SVDs, ensures that CLoQ is both computationally efficient and highly effective, particularly at ultra-low bit-widths. Our extensive experiments on multiple benchmark datasets demonstrate that CLoQ consistently outperforms existing LoRA-based methods for quantized models, such as QLoRA, in tasks requiring fine-grained precision. The results underscore the potential of CLoQ to enhance the performance of quantized models across a variety of downstream applications, including those that demand high accuracy, like arithmetic reasoning tasks. The simplicity and efficiency of CLoQ make it a promising approach for fine-tuning large-scale quantized LLMs in resource-constrained environments. Future work could further investigate the theoretical implications of different decompositions of the adapter matrices in CLoQ, and how these variations influence the performance of subsequent fine-tuning.

## 7 Acknowledgments

This work was partially supported by NSF grants DMS2208126, DMS-2110836, IIS-2110546, SUNY-IBM AI Research Alliance Grant, UAlbany-IBM CEAIS grant, and a start-up grant from SUNY Albany. We would also like to thank SUNY Albany for providing access to the Nvidia DGX Cloud.

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

# Appendix

# A    Experimental Details

## A.1    Language modeling

To study the capability of CLoQ, we fine-tune quantized models on the WikiText-2 training set and measure perplexity on the validation set. The hyper-parameters used for fine-tuning are provided in Table 11 and Table 12. We evaluate the models on the validation set at each epoch and report the lowest achieved perplexity.

## A.2    Arithmetic reasoning

### Single task (GSM8K)

To assess CLoQ's arithmetic reasoning capability, we fine-tune quantized models using the GSM8K training set and evaluate their accuracy on the test set. The hyperparameters used for fine-tuning are detailed in Table 11 and Table 12. Model performance is evaluated at each epoch on the test set, and we report the highest recorded accuracy.

### Multiple task

Following the framework proposed by Hu et al. (2023), we adopt a more integrated approach by training a single model across multiple tasks. Specifically, we fine-tune Llama2-7B and Llama2-13B on Math10K, a dataset that aggregates training samples from GSM8K, MAWPS, MAWPS-single, and AQuA. After finetuning, the models are tested on the evaluation sets of AQuA, GSM8K, MAWPS, and SVAMP. The hyper-parameters used for fine-tuning are detailed in Table 11 and Table 12. Moreover, instead of conducting evaluations at every epoch, we assess model performance only after the final epoch.

## A.3    Commonsense reasoning

To evaluate the commonsense reasoning capabilities of CLoQ, we consider eight key benchmark tasks: BoolQ, PIQA, SIQA, HellaSwag, WinoGrande, ARC-e, ARC-c, and OBQA. We adopt the framework proposed by Hu et al. (2023) and fine-tune a single model across all these tasks instead of training separate models. We fine-tune Llama2-7B and Llama2-13B on the merged training set and measure accuracy on the corresponding test sets. The hyper-parameters used for fine-tuning are detailed in Table 11 and Table 12. For evaluation, we forgo per-epoch assessments and instead report the final model's performance after the last epoch.

Table 11: Hyper-parameter for the finetuning of Llama2.

| Hyper-parameter | WikiText-2 | GSM8K | Arithmetic reasoning | Commonsense reasoning |
|---|---|---|---|---|
| Optimizer | AdamW | | AdamW | |
| Weight decay | 0.1 | | 1.0 | |
| LR scheduler | cosine | | linear | |
| Warmup ratio | 3% | | 10% | |
| Epochs | 3 | 6 | 3 | |
| Batch size | 64 | 32 | 16 | |
| Max sequence length | 1024 | 512 | 512 | |

Table 12: Best learning rate for Llama2-7B and Llama2-13B on the WikiText-2, GSM8K, and multiple Arithmetic Reasoning tasks.

| Group size | Task | Llama2-7B | | | Llama2-13B | | |
|---|---|---|---|---|---|---|---|
| | | 4 Bits | 3 Bits | 2 Bits | 4 Bits | 3 Bits | 2 Bits |
| 64 | WikiText-2 | 7e-4 | 7e-4 | 6e-4 | 2e-4 | 4e-4 | 4e-4 |
| | GSM8K | 3e-4 | 3e-4 | 3e-4 | 4e-4 | 4e-4 | 3e-4 |
| | Arithmetic reasoning | 5e-4 | 9e-4 | 4e-4 | 2e-4 | 4e-4 | 5e-4 |
| | Commonsense reasoning | 8e-5 | 1e-4 | 4e-5 | 5e-5 | 7e-5 | 5e-5 |
| 128 | WikiText-2 | - | 7e-4 | 5e-4 | - | 2e-4 | 5e-4 |
| | GSM8K | - | 7e-4 | 5e-4 | - | 5e-4 | 4e-4 |
| | Arithmetic reasoning | - | 7e-4 | 6e-4 | - | 3e-4 | 5e-4 |
| per-channel | WikiText-2 | 7e-4 | 4e-4 | 4e-4 | 2e-4 | 2e-4 | 5e-4 |
| | GSM8K | 4e-4 | 4e-4 | 4e-4 | 4e-4 | 5e-4 | 5e-4 |
| | Arithmetic reasoning | 6e-4 | 8e-4 | 3e-4 | 2e-4 | 4e-4 | 5e-4 |

