# OpenReview forum: "CLoQ: Enhancing Fine-Tuning of Quantized LLMs via Calibrated LoRA Initialization"
_TMLR — Accepted by TMLR_

### Review · Reviewer_JtkH · 2025-06-06

**Summary Of Contributions:**

The paper focuses on parameter efficient fine-tuning of quantized LLMs using LoRA. The specific problem considered is how LoRA should be initialized in this setting. The proposed solution, termed CLoQ, contains two stage. In the first stage, a small calibration dataset is used for pretrained model quantization using an existing gradient-free method for solving $\min_Q\Vert X(Q-W)\Vert_F^2$, where $Q$ is the resulting quantized weight, $W$ is the pretrained weight, and $X$ is the activation when feeding the calibration data. In the second stage, LoRA initialization is obtained by solving $\min_{A,B}\Vert X(AB^\top + Q - W)\Vert_F^2$ using two SVD. Experiments show that CLoQ achieves better performance on several reasoning tasks, especially in the low-bit case.

**Audience:**

Yes

**Broader Impact Concerns:**

N/A. There are no immediate ethical concerns.

**Claims And Evidence:**

Yes

**Requested Changes:**

I would support the acceptance of the paper if the following adjustments are made.

On the writing side,

1. In the second paragraph of contribution, the paper says "Our CLoQ method requires no back-propagation, making it a highly efficient approach for fine-tuning quantized models". This could be misleading as CLoQ is only an initialization scheme. Back-propagation is still required for fine-tuning. Changing it to something like "making it a highly efficient approach for LoRA initialization" makes more sense.

2. It is better to introduce that W is the pretrained weights already in equation (1).

3. Although CLoQ requires just 2 SVD while LoftQ requires 5, CLoQ also need to compute the costly matrix inversion or pseudo-inversion. This is not discussed at all in the paper. The experiments later show that CLoQ is a little slower than LoftQ, which I guess comes from the matrix inversion. When comparing with LoftQ at the beginning of page 6, it is not fair to just say the advantage of CLoQ. Actually, LoftQ is faster.

4. The reference for Mistral 7b repeats twice (see page 13).

On the experimental side, it is better to run using different random seeds and report mean, std. This should be done at least for just 1 task, say Table 4, to show that the results are consistent. If the authors do not have time for these, I suggest report the current used random seed with details on how the results can be reproduced.

There are some missing related works studying initialization of LoRA, see [1,2,3]. Although these are not originally designed for quantized fine-tuning, it is also good to include them in the related works. Specifically, [2] also considers using SVD for initialization.

Reference:

[1] OLoRA: Orthonormal low-rank adaptation of large language models. 2024.

[2] PiSSA: Principal singular values and singular vectors adaptation of large language models. NeurIPS, 2024.

[3] On the crucial role of initialization for matrix factorization. ICLR, 2025.

**Strengths And Weaknesses:**

The paper is clearly written. The claims are supported by evidence. The results look good. The topic is interesting and important for the community. The proposed algorithm might look like a straightforward and easy extension of existing methods, but I think this should not affect the overall contributions and should not be listed as weakness. My actual concerns are

1. Some parts are not written very rigorously. See requested changes below for more details.

2. I think all experimental results are just one single run? Correct me if I am wrong. It is not clear whether the improvement just comes from randomness. It is better to average over several random seeds and also report standard deviation, at least for some examples that are not too heavy to run.

Other questions:

1. In Figure 1, some quantized methods achieve even better performance than the baseline LoRA. What is the reason? Is it just some random observation?

2. In Table 6, although each one is a valid solution of equation (4), the performance is significantly different (acc 1.6 v.s. 33.7). What could be the cause of this drastically different behavior?

---

> ### Author Response · Authors · 2025-06-29
>
> Thanks for your constructive comments. All revisions are highlighted in read text in the updated manuscript.
>
> 1. Paper presentation: We have revised the paper according to your suggestions as described in the requested changes.
>
> 2. Experiments: Yes, all results are for a single run. In response to your request, we reported the mean accuracy and std for CLoQ over 5 random runs in Table 4, and the finding is consistent that CLoQ significantly outperforms the competitors at 2-bit for arithmetic reasoning tasks.
>
> To answer your questions:
>
> 1. LoRA involves solving a complex non-convex optimization problem over a low-dimensional manifold, making its performance highly sensitive to initialization. The default initialization scheme (using a Gaussian matrix A and zero matrix B) has been shown to be suboptimal. Recent studies have proposed improved initialization strategies. Therefore, with a well-chosen initialization, quantized fine-tuning can sometimes outperform the standard LoRA baseline, particularly in the 4-bit regime.
>
> 2. While we do not yet have a rigorous theoretical understanding of how different initialization schemes affect performance, we believe the observed differences are closely tied to their impact on the subsequent training dynamics. In particular, proper initialization can guide optimization toward much better solutions. We plan to investigate this further and develop a more comprehensive understanding in future work.

---

### Review · Reviewer_koA6 · 2025-06-13

**Summary Of Contributions:**

The authors propose a new algorithm for initializing LoRA parameters for quantized LLM models. Their main idea is to minimize the quantization errors with the LoRA parameters using a small calibration set. Their method is efficient to implement and only requires SVD and no backpropagation. Experiments show that their method provides better finetuning results with LoRA when compared against several state-of-art LoRA parameter initialization methods.

**Audience:**

Yes

**Claims And Evidence:**

Yes

**Requested Changes:**

- The current paper is already quite polished. I would suggest the authors address the questions I raised in the weaknesses section above.

**Strengths And Weaknesses:**

Strengths:
- The main idea of this paper is to initialize the LoRA parameters by minimizing the errors introduced during quantization. The method relies only on SVD and does not require backpropagation to initialize the parameters. The paper is also very clearly written.
- Extensive experiments using Llama2 and LLama3 show improved finetuning performance on many benchmark tasks on arithmetic and commonsense reasoning, compared to competing methods like QLoRA, LoftQ, and ApiQ.
- The authors also provide good ablation studies on size of calibration sets required for the initialization using SVD, and LoRA initialization.

Weaknesses:
- The proposed algorithm works best for very low-bit quantization like 2-bit. For many tasks this quantization is too aggressive and there are significant drops in performance for all finetuning methods. This could limit the number of practical use cases for the algorithm.
- For Table 3, why is the 16-bit LoRA performance lower than 4-bit and 3-bit finetuning? Is it related to overfitting?
- Although the authors explain the differences of their method compared to ApiQ, which is the improved efficiency without the use of backpropagation. However this does not explain the difference between the two methods in terms of finetuning performance. We see the proposed CloQ method is better on the arithmetic reasoning tasks in Table 3, but ApiQ is better (especially for 3 and 4 bit quantizations) for commonsense reasoning tasks in Table 4. Do the authors have any intuitions on the differences in performance for these two initialization methods?

---

> ### Author Response · Authors · 2025-06-29
>
> Thanks for you helpful comments. Our responses to the weaknesses are as follows:
>
> 1. Existing state-of-the-art methods already achieve strong fine-tuning performance on 4-bit quantized models, and the accuracy gains at this precision level have largely saturated. Therefore, the main focus of CLoQ is on ultra low-bit regimes, such as 2-bit, where there is still significant room for improvement.
>
> 2. LoRA involves solving a complex non-convex optimization problem over a low-dimensional manifold, making its performance highly sensitive to initialization. The default initialization scheme (using a Gaussian matrix A and zero matrix B) has been shown to be suboptimal. Recent studies have proposed improved initialization strategies. Therefore, with a well-chosen initialization, quantized fine-tuning can sometimes outperform the standard LoRA baseline, particularly in the 4-bit regime.
>
> 3. Commonsense reasoning is generally a simpler task compared to arithmetic reasoning, and as discussed in (1), existing methods already perform well at higher bit-widths. On this task, our method does not show a significant advantage and is occasionally slightly outperformed by existing approaches at 4-bit or even 3-bit.

---

### Review · Reviewer_xv2J · 2025-06-16

**Summary Of Contributions:**

In this paper, the authors introduced calibrated LoRA initialization for quantized LLMs to minimize the layer-wise difference between the original LLM and its quantized version with LoRA components during initialization

**Audience:**

Yes

**Broader Impact Concerns:**

No concerns.

**Claims And Evidence:**

Yes

**Requested Changes:**

**1.** Fine-tuning results with mixing more diverse datasets. For example, mixing WikiText-2, Math10K, and commonsense reasoning.

**2.** Show the performance of the proposed method on Larger models like 30B or 70B models. Most people use quantized LoRA to tune very large models.

**3.** Ablation study with different sequence lengths.

**4.** The presentation of Tables 1 and 2 could be improved by separating the two tasks since these two tasks are fine-tuned on different datasets, and they'd better be separated.

**1.** is critical to my recommendation. **2.** is better to have. **3.** and **4.** are less important but straightforward to address.

**Strengths And Weaknesses:**

**Strength:**
This paper enhances existing LoRA initialization methods for quantized LLMs by incorporating features into the reconstruction error computation. This natural extension provides a clear and foreseeable path for further enhancing LoRA initialization quality in quantized LLMs, like LoftQ. In addition, the proposed method maintains similar computational costs compared to previous methods, and the problem can still be solved analytically.

**Weakness:**
It is reasonable that adding feature maps to the reconstruction problem improves the performance. In the paper, different tasks use different datasets for fine-tuning, and this setting could be the most beneficial one since the samples are concentrated in a single direction. Under this setting, it could be easier to find a good LoRA initialization fitting the target task. The author could also consider a more challenging setting, where diverse datasets are mixed together for fine-tuning. Such a result would strengthen the paper and better understand the proposed method.

---

> ### Author Response · Authors · 2025-06-29
>
> Thanks for the help comments. The responses to the requested changes are summarized below, and the corresponding modifications are marked in red in the revised file.
>
> 1.We obtained fine-tuning results on a mixed dataset and compared CLoQ with LoftQ (Table 6). Specifically, we fine-tuned on a dataset combining Math10K with Commonsense samples and evaluated performance on four arithmetic reasoning tasks. We observed that mixing datasets leads to a degradation in overall accuracy. Nevertheless, under this setting, CLoQ still outperforms the competitor.
>
> 2. Thank you for your suggestions. Due to time constraints, we were unable to fine-tune very large models. However, we note that the other methods in our comparison also did not include results for 30B or 70B models. We believe that CLoQ's superior performance would likely extend to these larger models as well.
>
> 3. Based on your suggestions, we've added the finetuning results for different sequence lengths in Table 9.
>
> 4. The current version already includes more than ten tables, and splitting Tables 1 and 2 would further increase the number of tables. Moreover, we find that Tables 1 and 2 already present the results clearly and concisely in their current form. Therefore, we have decided not to make changes at this time.

---

### Decision · Action_Editor_sC6r · 2025-08-05

**Recommendation:** Accept as is

**Audience:**

Yes

**Audience Explanation:**

Yes. Reviewers indicate clear interest, highlighting the method's practical advantages, especially for audiences working with quantized LLMs and finetuning them on downstream tasks. The idea is also very simple and easy-to-implement, so it is widely applicable.

**Claims And Evidence:**

Yes

**Claims Explanation:**

Yes. Reviewers agree that the proposed approach is justified by a proof, and the paper supports its claims with experimental evaluations. The proposed initialization method's effectiveness, especially in low-bit settings, is convincingly demonstrated via experiments. Reviewers do suggest improvements, such as running experiments over multiple seeds, but overall find the evidence clear.